# Metabolic Dysfunction-Associated Steatotic Liver Disease: From a Very Low-Density Lipoprotein Perspective

**DOI:** 10.3390/biom15070990

**Published:** 2025-07-11

**Authors:** Yan Chen, Kaiwen Lei, Yanglong Liu, Jianshen Liu, Kunhua Wei, Jiao Guo, Zhengquan Su

**Affiliations:** 1Guangdong Engineering Research Center of Natural Products and New Drugs, Guangdong Provincial University Engineering Technology Research Center of Natural Products and Drugs, Guangdong Pharmaceutical University, Guangzhou 510006, China; 2112340214@stu.gdpu.edu.cn (Y.C.); 2112342038@stu.gdpu.edu.cn (K.L.); 2112342093@stu.gdpu.edu.cn (Y.L.); 2112340238@stu.gdpu.edu.cn (J.L.); 2Guangdong Metabolic Disease Research Center of Integrated Chinese and Western Medicine, Key Laboratory of Glucolipid Metabolic Disorder, Ministry of Education of China, Guangdong TCM Key Laboratory for Metabolic Diseases, Guangdong Pharmaceutical University, Guangzhou 510006, China; 3Key Laboratory of State Administration of Traditional Chinese Medicine for Production & Development of Cantonese Medicinal Materials, Guangzhou Comprehensive Experimental Station of National Industrial Technology System for Chinese Materia Medica, Guangdong Engineering Research Center of Good Agricultural Practice & Comprehensive Development for Cantonese Medicinal Materials, School of Chinese Materia Medica, Guangdong Pharmaceutical University, Guangzhou 510006, China; weikunhua@gdpu.edu.cn

**Keywords:** metabolic dysfunction-associated steatotic liver disease, lipoproteins, liver, triglycerides, very low-density lipoprotein

## Abstract

Metabolic dysfunction-associated steatotic liver disease (MASLD) is characterized by excessive accumulation of triglycerides and other lipids within liver cells and is closely associated with cardiovascular disease and metabolic syndrome. Very low-density lipoprotein (VLDL) is a lipoprotein synthesized and secreted by the liver and is primarily responsible for transporting triglycerides from the liver to peripheral tissues. Therefore, there is a strong association between MASLD and VLDL. Studies have found that excess production and abnormal metabolism of VLDL can lead to elevated blood triglyceride levels, which in turn promote fat deposition in the liver, leading to MASLD. During the pathophysiological process of MASLD, adipokines and inflammatory mediators secreted by adipose tissue can affect the metabolic network of the liver, further aggravating VLDL metabolic disorders. This paper reviews the effects of VLDL synthesis and metabolism on the development of MASLD, including the changes in VLDL structure and composition, the biosynthesis of VLDL, and the mechanism of underlying VLDL-associated damage, in an attempt to elucidate the intricate crosstalk between MASLD and VLDL, in order to provide new perspectives and methods for the prevention and treatment of related diseases.

## 1. Introduction

The liver is a central metabolic organ, which plays an important role in maintaining lipid metabolism. Because it is the primary site of fatty acid synthesis, metabolism, and transport, healthy livers typically do not accumulate large amounts of neutral fats. However, when fatty acid intake, de novo synthesis, fatty acid oxidation, and lipid export, that is, any link in the liver lipid homeostasis, is disturbed, it can promote the development of MASLD.

The liver takes up free fatty acids (FFAs) from peripheral tissues through specific fatty acid transporters such as Fatty Acid Transport Protein 2 (FATP2), Fatty Acid Transport Protein 5 (FATP5), and Collagen Type I Receptor (CD36). Among them, FATP2, a fatty acid transporter, is highly expressed in the liver and can increase the uptake of LCFAs (long-chain fatty acids) but does not affect the transport of glucose and short-chain fatty acids [1]. Falcon, A. et al. [2] constructed a FATP2 gene knockout mouse model to overexpress or knock down FATP2 in cultured liver cells. Using techniques such as radioisotope labeling and mass spectrometry, they investigated the effects of FATP2 deficiency on liver lipid metabolism. The results showed that FATP2 not only transports long-chain fatty acids in the liver but also acts as an acyl-CoA synthetase in peroxisomes, participating in the activation and metabolism of fatty acids. Therefore, FATP2′s dual role is essential for maintaining liver lipid homeostasis, and its deficiency or dysfunction can lead to various issues, such as a fatty liver and metabolic syndrome. These fatty acids can be derived from the diet, the breakdown of fat tissue, or metabolites from other tissues. Fatty acid intake is the starting point of liver lipid metabolism, and its efficiency directly affects the subsequent metabolic process. If the intake of fatty acids is too much, beyond the metabolic capacity of the liver, it will cause fatty acids to accumulate in the liver and then trigger MASLD.

De novo lipogenesis (DNL) allows the liver to synthesize new fatty acids from acetyl coenzyme A. Acetyl-CoA carboxylase (ACC) converts acetyl-CoA into malonyl-CoA. Subsequently, fatty acid synthase (FASN) gradually converts malonyl-CoA and acetyl-CoA into long-chain saturated fatty acids, such as palmitic acid. Saturated fatty acids, like C16:0-CoA, are converted into monounsaturated fatty acids, such as C16:1-CoA, by the action of stearyl-CoA desaturase (SCD). Finally, fatty acid elongase (the LOVL family of enzymes) extends short- and medium-chain fatty acids into long-chain fatty acids. The glycerol-3-phosphate acyltransferase (GPAT) binds glycerol-3-phosphate (glycerol-3-phosphate) with acyl-CoA (acyl-CoA), forming lysophosphatidic acid. Subsequently, the 1-acyl-glycerol-3-phosphate O-acyltransferase (AGPAT) catalyzes the reaction of lysophosphatidic acid with another acyl-CoA. The combination of phospholipid acid is catalyzed by phospholipid acid phosphatase (PAP), which dephosphorates the phospholipid acid to form diacylglycerol. Diacylglycerol acyltransferase 1 (DGAT1) then combines acyl-CoA with diacylglycerol to produce triglycerides. Finally, microsomal triglyceride transfer protein (MTP) facilitates the transfer of triglycerides and cholesterol esters from the endoplasmic reticulum to lipoproteins. Among these particles, VLDL particles are eventually formed and secreted to the extracellular space. Problems with DNL may lead to steatohepatitis. Because saturated fatty acids can cause inflammation and apoptosis [3]. DNL is regulated by two key transcription factors, Sterol Regulatory Element Binding Protein 1C (SREBP-1C) and Carbohydrate Response Element Binding Protein (ChREBP) [4], whose increase leads to increased fatty acid synthesis, which also increases lipid accumulation. When DNL is overactive, triglyceride synthesis in the liver increases. If these triglycerides are not oxidized in time or exported via VLDL, they can accumulate in the liver, leading to a fatty liver.

Fatty acid oxidation (FAO) occurs primarily in mitochondria, is controlled by PPARa, and provides an energy source for ATP [5]. The increase in FAO can reduce the storage of fatty acids in the liver, thus affecting the assembly and secretion of VLDL. When the expression of fatty acid oxidase in the liver is increased, more fatty acids are broken down into energy, and fewer triglycerides are available in the liver for VLDL assembly, thus inhibiting VLDL production. On the contrary, if FAO is inhibited, the accumulation of triglycerides in the liver will increase, thus promoting the synthesis and secretion of VLDL [6]. Apart from fatty acid oxidation, triglyceride export is the only way to lower liver lipids [7], but it is hydrophobic and therefore needs to be packaged in VLDL particles for export from the liver [8]. Each VLDL particle has an ApoB100 molecule, which is necessary for VLDL export. ApoB100 requires re-microsomal triglyceride transfer protein (MTTP) catalysis [9], so ApoB100 and MTTP are key to VLDL secretion and the maintenance of lipid homeostasis. When VLDL secretion is impaired (such as ApoB100 or MTTP dysfunction), triglycerides cannot be efficiently exported, causing them to accumulate in the liver, further aggravating the fatty liver. It is evident that VLDL plays a pivotal physiological role in lipid metabolism and is intricately linked to numerous diseases, including cardiovascular ailments and metabolic syndrome.

In Cushing’s syndrome, total cholesterol and triglycerides, as well as VLDL triglycerides, are elevated, presenting as lipid and lipoprotein abnormalities. The average concentrations of LDL and HDL are also elevated, resulting in hypercholesterolemia. The main mechanism for the increase in VLDL and total triglyceride levels is the excessive production of VLDL but unchanged clearance [10,11].

In addition to contributing to the progression of atherosclerosis, VLDL may also be directly related to the adrenal hormone aldosterone, promoting the production of aldosterone and causing hypertension. Physiological concentrations of VLDL, a member of the lipoprotein family, stimulate the production of aldosterone in adrenocortical cells through the increased expression of the acute regulatory protein (StAR) and aldosterol synthase (CYP11B2), possibly mediated by a calcium-initiated signaling cascade or through PKA, ERK1/2, and Janus Kinase 2 (JAK-2) signaling cascades [12,13].

In addition to the above diseases closely related to VLDL, most importantly, it is also closely related to metabolic syndrome and a new challenge facing modern liver disease: metabolic dysfunction-associated steatotic liver disease (MASLD). MASLD involves a variety of disease processes that can develop from initial simple hepatic steatosis to severe nonalcoholic steatohepatitis (NASH), which can further progress to liver fibrosis, cirrhosis, and even hepatocellular carcinoma (HCC) [14].

A recent meta-analysis estimated the global prevalence at 30% [15]. Its prevalence is expected to increase further as obesity and diabetes epidemics develop [16]. Dyslipidemia in MASLD is most pronounced in the MAFL or mild steatohepatitis stage, which is characterized by significant hepatic steatosis, and is attenuated in the more advanced fibrotic stage. The change associated with the MASLD stage may be closely related to dynamic alterations in VLDL production and metabolism. In MASLD patients, the overproduction of VLDL-TG is a notable feature. VLDL-TG serves as a precursor of small, dense low-density lipoprotein cholesterol (sdLDL-C), which is recognized as a sensitive biomarker for MASLD. Since VLDL-TG is overproduced in MASLD patients, its composition may have a more direct association with MASLD than sdLDL-C or plasma TG levels. In the context of MASLD, the overproduction of VLDL-TG can be attributed to the increased activity of enzymes involved in hepatic fatty acid synthesis and enhanced synthesis of apolipoprotein B, leading to greater VLDL-TG secretion by the liver. The elevated levels of VLDL-TG in the bloodstream, when subjected to the action of lipoprotein lipase, contribute to the formation of sdLDL. As a biomarker, sdLDL-C reflects this transformation process and is associated with an increased risk of cardiovascular disease in MASLD patients. Since VLDL-TG is a precursor of sdLDL and is overproduced in MASLD patients, the composition of VLDL may be more directly related to MASLD than sdLDL-C or plasma TG [17].

In recent years, the pathogenesis of MASLD has been widely accepted to involve the “second hit” or “multiple hits” theories, both of which include lipid accumulation. Most patients with MASLD will exhibit more or less some properties of metabolic syndrome, particularly atherogenic dyslipidemia characterized by insulin resistance or high plasma triglyceride concentrations. It is well known that unoxidized fatty acids (FAs) can be esterified into triglycerides (TGs), secreted from the liver as VLDL, and can reduce liver lipid content. When the liver’s lipid regulatory mechanisms become disrupted, TG levels may rise excessively, leading to the accumulation of fat in the liver. This state stems from differences in the regulation of lipid management processes. Lipid acquisition includes fatty acid uptake and de novo lipid synthesis. Subsequent clearance involves mitochondrial fatty acid oxidation and lipid secretion in the form of VLDL granules [18,19,20]. A fatty liver is formed when the fat accumulation in the liver exceeds 5% of the liver weight [21].

Liver steatosis is the first step in the development of MASLD and is characterized by a large accumulation of lipids in the liver, which may result from a variety of influencing factors. There are four mechanisms that lead to triglyceride accumulation in the liver, as shown in Figure 1 below: (1) increased dietary fat intake; (2) increased lipolysis in subcutaneous or visceral adipose tissue, or high de novo fat production in the liver, resulting in increased free fatty acid influx; (3) insufficient beta-oxidation of fatty acids in mitochondria; and (4) decreased secretory output of VLDL [22]. Therefore, when FFA oxidation or VLDL secretion cannot utilize the overloaded FFA, excessive FFA will be esterified into TGs and stored in lipid droplets, resulting in lipid accumulation and the fatty degeneration of hepatocytes, that is, hepatocyte steatosis and TG accumulation occurs when lipid input exceeds lipid utilization [23].

Previous studies have demonstrated that inflammatory bowel disease (IBD) mice induced by a normal diet (NC) or a high-fat diet (HF) in combination with sodium dextran sulfate (DSS) treatment exhibit significant down-regulation of microsomal triglyceride transfer protein (MTP) and apolipoprotein B (ApoB) levels via the hepatocyte nuclear factor 4α (HNF4α) pathway. This down-regulation inhibits the secretion and outflow of very low-density lipoprotein triglyceride (VLDL-TG) from hepatocytes, leading to excessive accumulation of TGs in the liver. This accumulation exacerbates the progression of MASLD [24].

In summary, it can be inferred that the dysregulated metabolism of VLDL significantly contributes to the initiation and progression of MASLD. Consequently, modulating the synthesis and metabolic pathways of VLDL may emerge as a crucial therapeutic strategy for MASLD management. Hence, the subsequent sections will delve deeper into elucidating the intricate relationship between the pathophysiological mechanisms underlying MASLD and the role of VLDL.

## 2. The Pathogenesis and Factors of MASLD from VLDL

In recent years, more and more studies have shown that MASLD is now the leading cause of incidence and mortality of liver-related diseases [25]. The “Two-hit” hypothesis is a classic hypothesis for the pathogenesis of MASLD. This hypothesis posits that MASLD originates from the accumulation of fat in the liver, often due to excessive nutrient intake, which disrupts the balance between the body’s storage and utilization of liver fat. The “Two-hit” hypothesis further suggests that obesity or diabetes mellitus can lead to fatty degeneration and an increase in free fatty acids in the liver. These conditions may trigger the production of lipotoxic agents, including lysophosphatidylcholine, lysophosphatidic acid, ceramide, and reactive oxygen species (ROS), within hepatocytes. These lipotoxic substances induce oxidative stress, which in turn causes further liver damage [26]. However, the “Two-hit” hypothesis cannot explain the occurrence and development of MASLD well, and more and more studies show that MASLD is the most common multi-factor disease, so the “Two-hit” theory has been proposed, and the combination of insulin resistance [27], adipokine secretion, mitochondrial dysfunction, oxidative stress, endoplasmic reticulum stress, genetics, and other factors [28] eventually leads to liver damage, leading to the occurrence of MASLD. VLDL plays an important role in these mechanisms. The pathogenesis of MASLD is describedin Figure 2, including all of the mechanisms affected and the assembly and secretion of VLDL through different pathways.

### 2.1. Lipid Metabolism

From the definition of MASLD (lipid accumulation in liver cells is at least 5% or more of liver weight), we know that lipid accumulation in the liver is one of its influencing factors. The four parts of liver lipid metabolism are interrelated and work together to maintain lipid homeostasis in the liver. A disturbance in either link can tip the balance, leading to an excessive accumulation of lipids in the liver, ultimately triggering MASLD: (1) Excessive intake of fatty acids can overwhelm the transport capacity of VLDL, leading to the accumulation of TGs in hepatocytes. This accumulation further exacerbates the development of MASLD. (2) Overload in DNL synthesis, driven by an increased activity of Sterol Regulatory Element Binding Protein-1C (SREBP-1C) and Carbohydrate Response Element Binding Protein (ChREBP), increases the burden on VLDL synthesis and contributes to lipid accumulation. (3) Reduced activity of peroxisome proliferator-activated receptor alpha (PPARα) results in decreased fatty acid oxidation, leading to increased lipid accumulation in the liver. (4) The abnormal function of microsomal triglyceride transfer protein (MTTP) or apolipoprotein B (ApoB) impairs VLDL secretion, preventing TGs from being effectively transported out of the liver. This impairment further contributes to lipid accumulation and the progression of MASLD. Dysregulation of any of the key regulators of these four processes, especially transcription factors, enzymes, or signaling pathways that control these metabolic processes, will interfere with hepatic lipid homeostasis, resulting in abnormal lipid deposition in the liver and progression to MASLD. Xiaohui Zhang et al. found that naringin reduced TG synthesis by down-regulating CD36 and ACC and up-regulating PPARα and CPT-1, which improved fatty acid beta oxidation and participated in improving liver lipid accumulation. Therefore, CD36 and PPAR-α may be specific targets of naringin in improving MASLD [29].

### 2.2. Insulin Resistance

In addition to its direct effects on lipid metabolism, VLDL metabolism is also profoundly influenced by endocrine regulation. Among these, insulin signaling is one of the key factors determining the balance between VLDL and MASLD. Insulin is a negative regulator of VLDL triglyceride and ApoB secretion [30]. High concentrations of insulin can inhibit the expression of MTTP [31,32] and reduce the translation of ApoB mRNA [33], thus inhibiting the production of VLDL. Insulin can inhibit VLDL production, and VLDL production increases and the clearance decreases in insulin resistance (IR). IR is closely related to MASLD and is a key factor in the pathogenesis of MASLD, so as many as 66.7% of patients with diabetes mellitus (T2DM) will have MASLD complications. The PI3K/AKT signaling pathway, essential for normal metabolism, becomes imbalanced and leads to the development of obesity and type 2 diabetes [34]. Insulin binds to its tyrosine receptor to the cell membrane, causing receptor autophosphorylation, which then phosphorylates its key substrates, insulin receptor substrate 1 and insulin receptor substrate 2. Insulin resistance selectively inhibits the hypoglycemic effects of insulin while continuing de novo lipogenesis through the activation of SREBP-1. SREBP-1 plays a key role in de novo fat synthesis [35]. The higher the SREBP-1 activity, the more the DNL becomes overloaded, thus increasing the burden on the VLDL. Sudha B Biddinger et al. [36] established a pure hepatic insulin resistance model in LIRKO mice (hepatic insulin receptor knockout mice) and found that the loss of insulin receptors in hepatocytes leads to a decrease in SREBP-1C, which regulates the expression of lipogenesis genes, and SREBP-2, which regulates the expression of cholesterol-producing enzymes and LDL receptors, affecting triglyceride accumulation. It was also suggested that insulin resistance inhibits SREBP-1C independently of the mechanism of LXR ligand production and that reduced expression leads to reduced VLDL-TG secretion. Quinn, W. J et al. [37] achieved the liver-specific knockout of Raptor (a key component of mTORC1) and TSC1 (a negative regulator of mTORC1) in mouse models by using AAV virus-mediated gene editing technology and found that mTORC1 is a key regulator of VLDL-TAG secretion and lipid homeostasis. Lipin-1 is a key regulator of lipid metabolism, and its activity is regulated by mTORC1. When there is excess nutrition, mTORC1 inhibits the transcriptional activity of Lipin-1, leading to the increased expression of fatty acid synthase (e.g., SREBP-1C), which promotes fat accumulation. Shuangshuang Zhang et al. [38] used both cellular and animal models to investigate lipid accumulation and VLDL secretion during the period. They concluded that Sphingosine Kinase 2 (SphK2) activates mTORC2 phosphorylation, which regulates Chaperone-Mediated Autophagy (CMA) and promotes VLDL secretion, thereby balancing liver lipid metabolism. The absence or inhibition of SphK2 leads to reduced VLDL secretion and increased lipid accumulation in the liver, which may be a potential mechanism for MASLD. Liver-specific receptor knockout mice demonstrated the importance of hepatic insulin signaling, with differential effects on VLDL secretion despite the absence of steatosis. It is well known that elevated FFA levels are closely associated with insulin resistance. The impairment of the anti-lipolytic action of insulin leads to increased lipolysis in adipocytes and increased circulating FFA, resulting in large amounts of FFA being sent to the liver and new lipogenesis, which in turn leads to steatosis and insulin resistance in muscle cells [39]. Park E et al. demonstrated that sodium salicylate of IκB Kinase β (IKKβ) prevents hepatic insulin resistance induced by the short-term elevation of FFA in female Wistar rats in the hyperinsulinemia-euglycemic clamp and tracer infusion experiments by the intravenous injection of different drugs [40]. Ceramide is also an important metabolite of sphingolipids and is closely related to insulin resistance [41]. Oversaturated FFA also reduces ApoB secretion through lipid-induced endoplasmic reticulum stress. This not only affects the secretion of VLDL but also many other diseases of MASLD.

### 2.3. Oxidative Stress

Oxidative stress (OS) occurs when reactive oxygen species (ROS) are overproduced and endogenous antioxidant molecules are deficient, which is also known as an imbalance in the production and elimination of oxygen-free radicals in the body [28,42]. Elevated OS leads to progressive hepatocyte death [43,44], promoting cirrhosis and liver cancer [45]. The interaction between gut-derived bacterial lipopolysaccharide (LPS) and toll-like receptor 4 promotes oxidative stress. LPS can damage hepatocytes and activate Kupffer cells to produce pro-inflammatory cytokines, which in turn release oxygen-free radicals. Increased LPS concentration can induce enhanced lipid peroxidation through oxidative stress damage [46]. OS triggers inflammation, increases the risk factors for atherosclerosis, and induces endothelial dysfunction to increase cardiovascular risk in patients with MASLD [47]. Jingda Li et al. [48] demonstrated in vivo and in vitro that hesperidin can upregulate antioxidants (SOD/GCLD/HO-1) by triggering the PI3K/AKT-Nrf2 pathway and reduce ROS overproduction and hepatotoxicity induced by OA, and the activation of NF-κB is known to improve oxidative stress in the liver through double verification in rat models of MASLD induced by HFD and HepG2 cells induced by oleic acid (OA). In MASLD induced by a high-fat diet, liver cells absorb and accumulate a significant amount of fatty acids, leading to the development of a fatty liver. Oleic acid, a monounsaturated fatty acid, can be absorbed by cells and esterified into triglycerides, resulting in the formation of lipid droplets and lipid accumulation within the cells. In this study, oleic acid was used to treat HepG2 cells, increasing intracellular lipid accumulation, which simulates the pathological condition of fatty acid overload, thereby mimicking the fatty degeneration of liver cells in MASLD [49,50]. The nuclear factor erythrocyte 2-related factor 2 (Nrf2), which is sensitive to oxidative reaction and activates transcription under oxidative stress, is related to VLDL secretion. Mice fed a high-fat diet or methionine- and choline-deficient diet showed decreased Nrf2 expression and oxidative stress in hepatocytes. When the translocation of Nrf2 is inhibited, VLDL maturation is impaired and lipid accumulation occurs, leading to hepatic steatosis and the further development of MASLD [51]. Oxidative stress can also affect VLDL: oxidative stress can not only damage mitochondria, lead to reduced oxidation of fatty acids, and increase the burden of VLDL synthesis, but it can also lead to insulin resistance through the activation of the JNK signal pathway and other pathways, affecting the normal secretion of VLDL. It has also been shown that glucosamine can reduce acute liver injury caused by alcohol by inhibiting oxidative stress and inflammation, improve the content of VLDL in mice, and promote lipid excretion [52].

### 2.4. Endoplasmic Reticulum Stress

Endoplasmic reticulum stress (ERS) refers to a condition where the normal function of the endoplasmic reticulum (ER) within cells is disrupted, typically due to imbalances in protein folding and processing. The ER can be triggered by various factors, such as hypoxia, nutrient deficiency, or the overload of unfolded protein. When the ER detects misfolded or unfolded proteins, it activates a series of signaling pathways. The Unfolded Protein Response (UPR) is one such pathway. This pathway enhances the production of molecular chaperones to assist in protein folding, thereby restoring the ER’s normal function and reducing related issues. However, if the stress persists for an extended period or is severe, the UPR may induce apoptosis to eliminate damaged proteins and cells. This dysfunction leads to the accumulation of misfolded proteins in the ER, triggering a series of adaptive and protective mechanisms to restore normal ER function. More and more studies show that severe endoplasmic reticulum stress and maladaptive UPR participate in the occurrence and development of liver pathology, including diabetes, insulin resistance, inflammatory diseases, nonalcoholic fatty liver, alcoholic fatty liver, viral hepatitis, liver ischemia reperfusion injury, liver fibrosis, and liver cancer [53,54]. After prolonged exposure to endoplasmic reticulum stress, liver VLDLR expression is upregulated through the activation of transcription factor 4 (ATF4) signaling, inducing hepatic steatosis [55]. Transcription factor XBP1 is a key regulator of UPR or endoplasmic reticulum stress, and the inhibition of its activity can treat MASLD or ALD [56]. Moreover, ER stress promotes the activation of SREBP-1C, a transcription factor that activates ACC1, ACC2, and FAS, and its activation promotes lipid synthesis [57]. Hélène L Kammoun et al. overexpressed glucose-regulated protein 78 (GRP78) in the liver of ob/ob mice via an adenovirus vector to inhibit insulin and endoplasmic reticulum stress-induced activation of SREBP-1C and reduce hepatic steatosis [58]. It can be seen from this experiment that endoplasmic reticulum stress can activate SREBP-1C, thus increasing DNL and aggravating the VLDL burden. Endoplasmic reticulum stress has also been shown to be closely related to NLRP3 inflammatory bodies (a 115 kDa cytoplasmic protein), which are involved in many pathological processes [59,60]. Endoplasmic reticulum stress can activate NLRP3, lead to the release of inflammatory factors, and affect VLDL metabolism. Xue Wu et al. [61] induced an MASLD rat model by HFD, treated with patchouli alcohol (PA), and found that PA can effectively alleviate the pathological process of MASLD by inhibiting protein kinase RNA-like endoplasmic reticulum kinase (PERK) and Inositol-Requiring Enzyme 1 (IRE1) and activating ATF6 to inhibit endoplasmic reticulum stress and increase ApoB-100 secretion, MTP activity, and VLDLR expression and restore VLDL synthesis and efflux.

### 2.5. Gut Microbiota

According to recent studies, the gut microbiota regulates carbohydrate, lipid, protein, and amino acid metabolism [62] and plays an important role in human health and various metabolic diseases (obesity, type 2 diabetes, dyslipidemia, MASLD, and atherosclerosis) [63]. However, stable and diverse communities can also be disrupted, resulting in a reduced abundance of beneficial bacteria, known as “gut microbiota disorders”, which is increasingly linked to the development of MASLD pathogenesis [64] and is a real target for MASLD disease treatment and intervention [65], with significant changes in the structure and composition of the gut microbiota in obese or MASLD patients and mice compared to healthy organisms [66]. Jorge Henao-Mejia et al. conducted experiments in transgenic mice deficient in inflammatory bodies and found that TLR4, TLR9, and other agonists flowed into the portal vein circulation to activate pro-inflammatory pathways, which aggravated liver steatosis, thus concluding that microbial structural changes were closely related to MASLD [67]. The gut microbiota can not only affect host energy metabolism and the immune function of the intestine and liver by changing intestinal metabolites, but it can also change the bile acid metabolism pathway to cause enterohepatic axis dysfunction and can finally be used as a screening marker for MASLD [68]. Imbalances of the gut microbiota will lead to the thinning of the intestinal barrier, resulting in LPS inflow, the activation of TLR4, and the induction of inflammatory factors (IL-6, TNF-α) leading to insulin resistance, further affecting VLDL secretion.

### 2.6. Bile Acid Metabolism

It is well known that there is a bi-directional interaction between intestinal microflora (IM) and bile acids (BAs), which affect intestinal homeostasis by controlling the size and composition of intestinal microflora, and bile acid pool composition is also affected by bacterial metabolism [69]. BA can alter the composition of microbial communities. By reducing the number of microorganisms that are sensitive to bile acid’s antibacterial effects and increasing those that rely on bile acids for growth, they influence the microbial community [70]. BAs are produced by the liver through the metabolism of cholesterol and have multiple biological effects related to MASLD. Additionally, BAs are also signaling molecules that coordinate metabolism and inflammation through the Farnesoid X Receptor (FXR) and G protein-coupled receptor 5 (TGR5) [71] and participate in maintaining intestinal homeostasis. A large number of observational studies have shown that obesity, T2D, MASLD, and NASH are all associated with changes in BA metabolism and bile acid pool composition [72,73,74]. BA can activate SREBP-1C via FXR to reduce DNL, thus reducing VLDL levels. Abnormal metabolism may also affect the assembly and secretion of VLDL. Yuanyuan Lei et al. used FXR knockout (FXR−/−) mice and normal mice fed with a choline-deficient, L-amino acid-defined diet (CDAHFD) to establish NASH mouse models and treated them with DSF. The results showed that (FXR−/−) mice with or without DSF did not have significant differences in liver steatosis, inflammation, or fibrosis levels, so it can be concluded that DSF is dependent on bile acid-induced FXR signaling to activate MAFH [75].

### 2.7. Mitochondrial Abnormalities

There is increasing evidence that mitochondrial metabolism is involved in the pathogenesis of MASLD to varying degrees. Because mitochondria play a key role in lipid metabolism, ROS and ATP production, and apoptosis [76], mitochondrial dysfunction is thought to be responsible for the severity of MASLD/MASH, and abnormal mitochondrial function has been identified as an important feature of a fatty liver in humans [77]. In patients with MASLD, mitochondrial dysfunction leads to the inhibition of fatty acid oxidation, resulting in lipid accumulation and hepatocyte damage [78]. Additionally, the imbalance in oxidative phosphorylation increases the production of ROS, leading to mitochondrial DNA damage, inflammation, and fibrosis. Furthermore, impaired insulin signaling exacerbates insulin resistance, while impaired autophagy further aggravates mitochondrial dysfunction. These changes collectively contribute to the pathological progression of MASLD [79]. From this, it can be seen that mitochondrial dysfunction will lead to the reduced oxidation of fatty acids, while TG accumulation aggravates the VLDL burden, and the increase in ROS will lead to the damage of MTTP and ApoB, affecting VLDL assembly and secretion, finally activating inflammatory pathways, aggravating insulin resistance, and affecting VLDL metabolism. The failure to remove damaged mitochondria or defective mitochondria in time can lead to cell necrosis and further progression to MASH [80,81]. Hao Zhou et al. [82] induced a mouse model of MASLD via a high-fat diet, and melatonin supplementation significantly alleviated mitochondrial dysfunction and MASLD. The mechanism is to restore mitochondrial autophagy by blocking the NR4A1/DNA-PKcs/p53 signaling pathway, thereby improving mitochondrial dysfunction in nonalcoholic fatty livers, manifested by enhanced ATP production, restored mitochondrial membrane potential, and improved mitochondrial respiratory function.

It can be seen that the pathogenesis of MASLD is complex, while VLDL is closely related to each of its pathophysiological processes. An important feature of liver cells is their ability to efficiently export excess triglycerides and other lipids to VLDL. If the rate of VLDL assembly and secretion is limited, the triglycerides are transferred to lipid droplets, and the degree of steatosis increases. The synthesis, assembly, and secretion mechanisms of VLDL are complex and regulated by different factors at different levels. Therefore, we will elaborate further below.

## 3. The Structure, Biosynthesis, and Influencing Factors of VLDL

### 3.1. Structural Composition of VLDL

Cholesterol and triglycerides are insoluble in water, so VLDL needs to be transported together with proteins, and lipoprotein is one of the key components involved in lipid metabolism in the liver, and it is also a macromolecule that transports lipids in plasma. Its basic structural characteristics are as follows: it is mainly composed of triglycerides and cholesterol esters, which constitute the core of VLDL, and is then surrounded by phospholipids and apolipoproteins, which can promote the formation and function of lipoproteins. The synthesis and regulation of cholesterol are vital for the secretion of VLDL. Cholesterol is mainly produced via the HMG-CoA reductase pathway, which is the key rate-limiting enzyme [83]. Cholesterol synthesis is primarily regulated through feedback mechanisms. When cellular cholesterol levels increase, HMGCR activity is inhibited, reducing cholesterol production. Conversely, when cholesterol levels drop, HMGCR activity rises, enhancing cholesterol synthesis. Additionally, the esterification of cholesterol significantly influences the assembly and secretion of VLDL. Sterol O-acyltransferases 1 and 2 (SOAT1 and SOAT2) play a crucial role in cholesterol esterification [84]. SOAT1 is predominantly located in the endoplasmic reticulum and is responsible for converting free cholesterol into cholesteryl esters, which are then packaged into VLDL particles [85,86]. SOAT2, on the other hand, is mainly found in the Golgi apparatus and participates in the further modification and transport of cholesteryl esters [87]. The activity of SOAT1/SOAT2 directly affects the formation of cholesteryl esters and the efficiency of VLDL secretion. Phospholipids are an important component of VLDL particles. Phospholipids are a class of amphiphilic molecules containing phosphate groups, which play a key role in the structure and function of VLDL particles. The main types of phospholipids include phosphatidylcholine (PC), phosphatidylethanolamine (PE), and sphingomyelin (SM). Choline is an essential nutrient required for the synthesis of PC, which is synthesized in the liver via two alternative pathways: the CDP-choline pathway and the methylation of phosphatidylethanolamine [88], and the CDP-choline pathway, also known as the Kennedy pathway, which is the primary route for PC synthesis in all mammalian tissues. PEMT is a liver-specific enzyme that converts PE to PC through a three-step methylation reaction, accounting for about 30% of PC synthesis in the liver [89]. In this pathway, choline is converted to CDP-choline, which then combines with diacylglycerol to form PC. PC is the major phospholipid of all types of lipoproteins in mammals and is a major component of cell and mitochondrial membranes. PC is the only phospholipid known to be essential for lipoprotein assembly and secretion, and it is also essential for VLDL secretion, which is not replaced by other phospholipids, and VLDL is responsible for transporting triglycerides to the outside of organs [90,91,92]. CDD-induced steatosis is characterized by high pathological and biochemical similarities to human fatty livers. High-density lipoprotein (HDL), low-density lipoprotein (LDL), VLDL, intermediate-density lipoprotein (IDL), chylomicron (CM), etc., are the main types of lipoproteins, and their densities range from high to low. These lipoproteins differ in the core lipids they carry and their surface proteins, apolipoproteins. The density of VLDL is very low, 0.93~1.006 g/mL, and the size is 30~80 nm. The core structural protein of VLDL is ApoB and contains other apolipoproteins such as ApoE, ApoC-I, ApoC-II, ApoC-III, ApoA-I, ApoA-II, ApoA-IV, and ApoA-V. The structure is shown in Figure 3 below [93]. In the case of ApoC-III, it is a small protein located on the surface of VLDL particles, a key regulator of triglyceride metabolism, which facilitates VLDL assembly and secretion [94]. Moreover, ApoC-III within cells promotes the secretion of VLDL, which may lead to excessive triglyceride secretion in the metabolic syndrome state, thereby affecting MASLD [95]. Studies have also shown that plasma ApoC-III prevents the lipolysis of VLDL mediated by LPL/ApoC-II, prolonging the retention time of VLDL in the plasma, which may lead to increased triglyceride levels and affect MASLD [96]. If ApoB, which provides structural stability to VLDL particles [97], of which ApoB-100 is a core structural protein produced by the liver, is changed, the composition of VLDL particles will change accordingly so that the amount of TG carried by each VLDL particle will vary greatly. ApoE, of course, has a molecular weight of 34 KDa and is a member of the amphipathic exchangeable apolipoprotein superfamily, one of the major determinants of lipid transport throughout the body and plays a key role in atherosclerosis and other metabolic syndrome diseases [98]. It is also a glycoprotein with two independent folding domains separated by an unstructured hinge. This unstructured hinge region is essential for proteins to perform their primary functions [99]. It can be confirmed through the amphiphilic α-helix and plays a role in lipid metabolism to stabilize lipid metabolism [100]. ApoE overexpression or accumulation may stimulate VLDL-TG production and impair VLDL lipolysis, leading to hypertriglyceridemia (HTG) [101]. Although HTG is more prevalent in MASLD, recent studies have shown that the accumulation of free cholesterol (FC) and cholesterol esters (CEs) also play a significant role in the pathogenesis of MASLD. The accumulation of FC has notable cytotoxic effects, including increasing ROS levels, impairing lysosomal function, disrupting lipid phagocytosis, and inducing cell apoptosis. Moreover, FC is directly linked to the inflammation and fibrosis associated with MASLD, possibly through the mechanism of increased toll-like receptor 4 (TLR4) signaling, which also triggers hepatocyte death by activating c-Jun N-terminal protein kinase 1 (JNK1). Hepatocyte death is considered a key trigger for the severity of MASLD, leading to subsequent inflammation, fibrosis, cirrhosis, and even hepatocellular carcinoma. Although the accumulation of CE is less significant in MASLD compared to FC, the accumulation of CE may promote liver inflammation and fibrosis by activating inflammatory signaling pathways such as NF-κB and JNK [102]. Therefore, while TG accumulation is a primary driver of steatosis, the roles of FC and CE in MASLD pathogenesis should not be overlooked. In addition to protein, VLDL mainly carries TG out of the liver, as well as cholesterol esters and phospholipids. The pathogenesis of NAFLD is characterized by the accumulation of triglycerides in the liver, and the defective export of VLDL to triglycerides leads to their accumulation in hepatocytes, which is a central mechanism in the development of hepatic steatosis [92]. The methionine- and choline-deficient diet (MCD) is a well-known experimental animal model for inducing fatty livers in rats [103], with severe hepatic steatosis similar to MASLD in humans but without weight gain [104]. In a mouse model of MASLD fed an MCD diet, choline deficiency greatly interferes with the synthesis or secretion of VLDL, which subsequently inhibits triglyceride transport out of hepatocytes, leading to fatty liver formation in rats [105].

### 3.2. Biogenesis and Assembly of VLDL

The biosynthesis of VLDL primarily occurs within the lumen of the endoplasmic reticulum (ER), which can be roughly divided into two main steps [106]. First, the formation of VLDL precursors is a critical initial step. ApoB-100 is synthesized in the ER lumen through co-translational translocation, and MTP (microsomal triglyceride transfer protein) mediates the covalent binding of lipids (TG, CE) to ApoB, forming lipid-deficient pre-VLDL particles (pre-VLDL). MTP not only plays a role in the assembly of VLDL precursors but is also essential for the stability of ApoB, Additionally, Hsp110 interacts with ApoB to inhibit its ubiquitination and degradation and ensure the stability of precursor particles [107]. VIGILIN [108], TIA1-like 1 (TIAL1), and Human antigen R (HUR) [109] affect ApoB synthesis and VLDL secretion by regulating the translation and splicing of ApoB mRNA, respectively. These original VLDL particles, known as pre-VLDL, then enter the next step of maturation.

In the first step, MTP is involved in VLDL precursor assembly, and lipids are added to ApoB in the ER in a co-translational translocation with the help of MTP to form original VLDL particles, an MTP-dependent process [110]. In the second step, neutral lipids (such as triglycerides) within the ER cavity are transferred to pre-VLDL via lipid transfer proteins (such as PLTP), forming mature TAG-rich VLDL particles. The nascent ApoB-100 is partially esterified into lipid-deficient virgin VLDL particles, which then fuse with triglyceride-rich particles or a large number of neutral lipids to form mature TAG-rich VLDL particles in the second step of VLDL formation. The liver experiences intermittent flow of free fatty acids from different sources, most of which are esterified with glycerol to form triglycerides. A major portion of de novo triglycerides are used to form liver-specific lipoproteins, VLDL, in the endoplasmic reticulum lumen [111]. This process may also involve other lipid transfer proteins, such as Phospholipase A2, Group XIIB (PLA2G12B), which regulates the lipid distribution between lipid droplets in the endoplasmic reticulum cavity and the newly formed VLDL, promoting the expansion of VLDL. VMP1 and TMEM41B assist VLDL in budding from the endoplasmic reticulum membrane into the endoplasmic reticulum cavity [112]. Mice with liver deficiency VMP1 also had impaired VLDL secretion and developed nonalcoholic steatohepatitis NASH [113].

Mature VLDL particles need to be transported from their synthesis site, the ER, to the next destination, the Golgi apparatus, where they further mature [114,115]. This transport process involves specialized transport vesicles known as VTVs, which are essential for the secretion of VLDL. The average diameter of VTVs is 100–120 nanometers, which is sufficient to encapsulate VLDL-sized particles, and each vesicle typically contains only one VLDL particle [116] and relies on COPII proteins (such as Sar1 and Sec23/24) for endoplasmic reticulum budding, with COPII proteins serving as the core driving force for VTV formation, and CideB acting as an auxiliary protein. Additionally, TANGO1 and TALI assist in recruiting VLDL at the endoplasmic reticulum exit site. It may allow VLDL to fuse with the ER–Golgi intermediate membrane, forming an exit channel. VTV is budding from the ER, and its formation is initiated by COPII proteins such as Sec13/31 and Sar1. CideB stabilizes the VTV membrane structure by binding to phospholipids, and its knockout results in a 70% reduction in VTV formation. Tiwari, S. et al. knocked out CideB by ER budding and siRNA knockout technology in vitro and found that the CideB blockade or knockout would significantly reduce the formation of VTV. Therefore, CideB protein is essential for the biogenesis of VLDL transport vesicles [117]. The Secretory Carrier Membrane Protein 24B (SEC24B) subunit selectively packages VLDL particles into VTV, ensuring the specific transport of lipid cargo. VTVs emerge from the endoplasmic reticulum and migrate to the cis face of the Golgi apparatus, where ApoE crosses the trans-Golgi network (TGN). Transmembrane proteins, such as Rab1, bind to the surface of VLDL. ApoB undergoes O-linked glycosylation (catalyzed by Golgi enzymes) and tyrosine phosphorylation, which enhances the stability of the particles. Mature VLDL is encapsulated into clathrin-coated vesicles (CCVs) within the trans-Golgi network (TGN) and transported to the plasma membrane via microtubules. The CCVs then fuse with the plasma membrane, and VLDL particles are released into the bloodstream through budding or lysis. Refer to Figure 4 for details.

### 3.3. Factors Affecting VLDL

Impaired VLDL secretion may be caused by mutations or defects of proteins (ApoB, MTTP, ApoE) essential for VLDL production, insulin action, the lack of the major phosphatidylcholine of the VLDL surface lipid, and possible fatty acid (FA) and non-esterified fatty acids (NEFA), or endoplasmic reticulum stress. Other factors such as phosphatidylcholine (PC) and cholesterol esters (CEs) also affect VLDL synthesis [118,119,120]. Refer to Figure 5 for details.

#### 3.3.1. Impact of MTP

Another essential factor for VLDL assembly and secretion is microsomal triglyceride transfer protein (MTP), which is not synthesized without MTP activity [119]. In addition to the intestine, heart, and liver, another major site of mammalian MTP expression, studies to date have shown that hepatocytes also express MTP. MTP is a heterodimeric protein consisting of a unique large α subunit (~97 KDa) and a multifunctional protein disulfide isomerase (PDI) β subunit (~55 KDa) [121,122]. MTP is involved in the assembly of triglyceride-rich chylomicrons in intestinal cells and VLDL in hepatocytes, which plays an important role in the transfer of neutral lipids to apolipoprotein ApoB at the early stage of lipoprotein assembly, and can promote the lipidation of nascent ApoB and the translocation of ApoB across the endoplasmic reticulum [123]. Recent studies have shown that the M subunit of MTP plays an important role in ApoB–lipoprotein assembly through protein–protein interactions with ApoB, binding ApoB through its N-terminal β-barrel domain [124], and the amino acid sequence 430–570 within the N-terminal 13% of ApoB is also critical for the MTP binding site [125]. Wei Fang et al. [126] established a large yellow croaker model or an in vitro liver cell model with lipid abnormality by treatment with HFD or OA. The analysis results showed that lipid overload could activate PKC δ through excessive ROS generation to prevent HNF4α from entering the nucleus, and a low HNF4α level in the nucleus inhibited MTP transcription, resulting in impaired hepatic VLDL secretion and abnormal TG accumulation, that is, lipid overload damaged hepatic VLDL secretion through the PKC δ-HNF4 α-MTP pathway mediated by oxidative stress. Studies have also shown that polymorphisms in the MTTP gene are associated with the risk of MASLD. Some researchers have found that the polymorphism of rs1800591 of the MTTP gene is strongly correlated with MASLD. In addition, Delilah Hendriks et al. [127] made wild-type organoids produce ApoB and MTTP mutants through CRISPR technology and confirmed that MTTP gene mutation can lead to lipid accumulation and affect MASLD.

#### 3.3.2. Effects of ApoB-100

In humans, ApoB exists in two forms, ApoB-100 and ApoB-48. It is a structurally amphiphilic protein with a size of 540 KDa, has a multi-domain structure, and is synthesized on membrane-bound polyribosomes. Both forms are derived from the same gene, converting the glutamine codon to a stop codon by the post-transcriptional modification of ApoB mRNA at codon 2153 [128], while apolipoprotein B-100 (ApoB-100) is the major protein component of plasma lipoproteins [129]. ApoB-100, synthesized by the liver, is a very large protein of 4536 amino acids containing multiple possible lipid-binding domains and is actually the only protein of LDL, a cholesterol ester-rich protein that is a metabolite of VLDL [130]. Many studies have shown that the co-expression of MTP and ApoB is necessary for ApoB assembly into triglyceride-rich lipoprotein particles [9,131]. Monosialyl ganglioside GM3 acts as an inhibitor of TG synthesis and secretion in cells, directly downregulating MTP synthesis levels. Furthermore, MTP is thought to be a target for inhibiting ApoB secretion and is essential for ApoB lipoprotein-containing liver secretion by mediating TG transfer to nascent ApoB [132]. Since GM3 inhibits ApoB secretion and MTP transcriptional activity, VLDL assembly and secretion in hepatocytes are largely controlled by newly synthesized ApoB [133]. AUP1 is an ER-related protein involved in the ERAD of unfolded proteins [134]. Recently, Jing Zhang et al. found that ApoB-100 was related to many proteins involved in the ERAD pathway, mainly molecular chaperones called heat shock proteins. Therefore, through AUP1 knockout experiments, it was found that ApoB-100-containing particles significantly increased the size of VLDL secreted by HepG2 cells, indicating that AUP1 is involved in the biogenesis, degradation, and lipidation of ApoB-100 in cells to test the assembly and secretion rate of VLDL particles. In addition, it can act as a key regulator of pre-secretory hepatic VLDL assembly [135]. Compared with healthy individuals, individuals with heterozygous inactivated ApoB mutations produce less VLDL and have a threefold increase in TG content in the liver, thus exacerbating the pathological process of MASLD.

#### 3.3.3. Effects of ApoE

ApoE, a 34.2 KDa glycoprotein, has three major natural isoforms in humans, ApoE2, ApoE3, and ApoE4, of which ApoE3 is the most common [136], and it is the main protein component of chylomicrons and VLDL. ApoE deficiency is characterized by reduced VLDL secretion and hepatic steatosis [137], so prevention and alleviation of hepatic steatosis are considered a function of VLDL secretion. ApoE plays a physiological role not only in plasma lipoprotein clearance but also in VLDL assembly and secretion in the liver, affecting lipoprotein particle size, and so on. If ApoE is absent, the VLDL particles produced become smaller, resulting in impaired VLDL-TG production, while TG synthesis in hepatocytes is not affected, resulting in lipid accumulation [138]. If ApoE is absent, the VLDL particles produced become smaller, resulting in impaired VLDL-TG production. This impairment is due to the reduced ability of these smaller particles to effectively transport triglycerides (TGs) out of the liver. Meanwhile, TG synthesis in hepatocytes remains unaffected, leading to increased lipid accumulation within the liver [127]. Additionally, insulin resistance has been shown to inhibit SREBP-1C independently of the mechanism of LXR ligand production. This reduced expression of SREBP-1C leads to decreased VLDL-TG secretion [127]. Regarding the formation of small, dense low-density lipoprotein (sdLDL), it is crucial to understand how triglyceride-rich very low-density lipoprotein (VLDL) is converted into cholesterol ester-rich low-density lipoprotein. During the lipoprotein metabolism process, VLDL particles in the blood are gradually converted into intermediate-density lipoprotein (IDL) by lipoprotein lipase (LPL) and eventually into low-density lipoprotein (LDL). Throughout this process, the composition of the LDL particles changes, with an increase in the relative content of cholesterol ester (CE). Cholesterol ester transfer protein (CETP) is a key regulatory protein primarily involved in the exchange process of CE and TG between plasma lipoproteins. CETP can transfer CE from HDL to lipoprotein particles rich in apolipoprotein B (ApoB), such as VLDL and LDL, while transferring TG from these particles to HDL. This process reduces the CE content in HDL particles by transferring CE to LDL and VLDL, increasing the CE content in LDL and VLDL particles. CETP alters the composition and function of these lipoproteins [139]. Although TG-enriched VLDL primarily consists of triglycerides, during metabolism, as triglycerides are gradually hydrolyzed, the cholesterol ester content in LDL particles increases. Therefore, from a metabolic perspective, the metabolic products of TG-enriched VLDL (i.e., LDL) may be rich in cholesterol esters. This does not mean that TG-enriched VLDL directly produces CE-enriched LDL; rather, it indirectly leads to an increase in cholesterol esters in LDL through the metabolic process. It has also been shown that transgenic mice lacking endogenous mouse ApoE have elevated VLDL-TG levels and increased VLDL-TG production by 50% when human ApoE3 with high plasma levels is chronically overexpressed in the liver. ApoE can also control VLDL-TG production by affecting a key step in the assembly cascade of VLDL particles [140]. Therefore, K Tsukamoto et al. [141] constructed recombinant adenoviruses of three human ApoE subtypes (E2, E3, E4) and injected them into the livers of ApoE-deficient mice. They used Triton WR1339 to inhibit the lipolysis of TG-rich lipoproteins and measured the secretion rate of VLDL-TG. The results showed that the increase in the VLDL-TG secretion rate was not related to the specific ApoE subtype expressed; all subtypes promoted the secretion of VLDL-TG. Studies have also shown that high levels of ApoE expression in the liver lead to a significant increase in VLDL triglyceride secretion [140,142]. In the MASLD state, the levels of LDLR and VLDLR in the liver rise. Based on this, some researchers [143] have developed DNNA-COP-Na-based lipid nanoparticles (LNPs) to deliver drugs to liver tissue through the ApoE-LDLR/VLDLR pathway, which is expected to improve MASLD.

#### 3.3.4. Impact of VLDLR

VLDLR is a member of the low-density lipoprotein receptor (LDLR) superfamily and is widely expressed in the brain, heart, etc., but is normally expressed at very low levels in the liver because it binds to ApoE, a triglyceride-rich lipoprotein such as chylomicron, and very low-density lipoprotein, resulting in reduced lipid secretion and increased lipid accumulation [120,144]. There have also been studies suggesting that increased abundance or upregulation of VLDLR leads to steatosis of hepatocytes in mice fed a high-fat diet that exacerbates the pathology of MASLD [145].

#### 3.3.5. Other Factors

Impaired secretion of VLDL is known to lead to the dysregulation of hepatic lipid homeostasis, thereby exacerbating the pathological progression of MASLD. Recently, several miRNAs, including miR-33, miR-122, miR-132, and miR-30c-5p, have been studied in lipid metabolism and have been shown to interact with key regulators of lipid synthesis [146,147,148,149]. Therefore Jing Zhang et al. [150] confirmed that miR-130b indirectly up-regulates MTP expression and promotes triglyceride mobilization through different in vitro culture models of HepG2 and Immortalized, thereby enhancing VLDL assembly and secretion in the liver. In addition to microRNAs (miRNAs), Long noncoding RNAs (lncRNAs) are an important class of ubiquitous genes involved in multiple biological functions [151], including lipid accumulation in the liver [152]. Some IncRNAs have been well documented to be associated with key transcription factors regulating cholesterol and triglycerides, such as LeXis [153], MeXis [154], and so on. Lingling Wang et al. found that lncRP11-675F6.3 and HK1 reduced MASLD induced by a high-fat diet by regulating VLDL-related protein and autophagy. lncRP11-675F6.3 may participate in the downstream of the mTOR signaling pathway and the regulatory network of hepatic triglyceride metabolism, interact with HK1, and provide a new target for the treatment of MASLD [155].

Phospholipids (PLs) on lipoprotein surface monolayers bind to ApoB at early stages of VLDL assembly in the liver, and PLs can also bind to ApoB at later stages of VLDL assembly [156,157]. It can be seen that PLs play an indispensable role in the formation and secretion of VLDL [91], so the PL composition of plasma may also affect lipoprotein metabolism. The major PL in mature VLDL are phosphatidylcholine [158] and sphingomyelin (SM), which make up approximately 20% of the phospholipids in human plasma proteins, two-thirds of which are present in LDL and VLDL [159], while 30% of PC in the liver is produced by phosphatidylethanolamine N-methyltransferase (PEMT) biosynthesis [160]. Yang Zhao et al. [89] conducted an experiment where mice deficient in PEMT and LDL receptors (Pemt (−/−)/Ldlr (−/−) mice) were fed a high-fat/high-cholesterol diet for 16 weeks. The results showed that the absence of PEMT led to a moderate reduction in PC production and a corresponding decrease in VLDL secretion in the liver of these mice. This highlights the critical role of PEMT in PC synthesis, particularly through the methylation of phosphatidylethanolamine to form PC. Since PC is a key component of VLDL particles, reduced PC levels directly impact VLDL assembly and secretion.

Studies have shown that non-esterified fatty acids (NEFAs) can act as a signal molecule involved in the gene expression of lipid metabolism [161]; plasma NEFA is the main substrate for hepatic triglyceride esterification and secretion, providing most of the fatty acids secreted by the liver in VLDL particles, and the proportion is more than 60% of total hepatic triglycerides secreted in the form of VLDL [162,163]. Many early studies have shown that increased plasma NEFA levels lead to increased VLDL–triglyceride secretion, so plasma NEFA pools are the only source of FAs for VLDL–triglyceride synthesis [164,165]. However, high plasma NEFA levels or concentrations do not necessarily significantly contribute to increased VLDL–triglyceride secretion [166]. Morten B Krag et al. [167] investigated the relationship between GH, FFA, and VLDL-TG in nine healthy men in a randomized, double-blind, placebo-controlled trial. It can be seen from the results that an increase in FFA flux is not necessarily related to an increase in VLDL-TG production in healthy men. Lei Liu et al. [168] verified in vitro experiments of bovine hepatocytes that high concentrations of NEFA could significantly inhibit the expression of ApoB-100, ApoE, MTP, and LDLR, leading to the synthesis and assembly of VLDL and inducing the accumulation of TG in bovine hepatocytes.

## 4. Targeted VLDL Delay the Development of MASLD

### 4.1. Preclinical Studies

MASLD has become a major cause of chronic liver disease worldwide, characterized by liver fat accumulation, lipid metabolism disorders, and especially VLDL secretion abnormalities. VLDL plays a key role in liver lipid transport, and its dysregulation not only leads to liver fat accumulation but also leads to complex dyslipidemia, which further aggravates the pathological process of MASLD. The following mainly introduces the targeted therapy strategies for VLDL in the direction of MASLD.

#### 4.1.1. Effect of TM6SF2 Gene Knockout on VLDL Lipidation

Fei Luo et al. [169] constructed a TM6SF2 (−/−) (TM6SF2 gene knockout) rat model by using gene editing technology CRISPR-Cas9 and found that TM6SF2 plays a role in ER and Golgi intermediate (ERGIC) and promotes large lipidization of ApoB-rich lipoproteins, thereby preventing the development of fatty liver disease. These results suggest that TM6SF2 is important for maintaining the homeostasis of hepatic lipid metabolism. Newberry EP (Elizabeth P Newberry) et al. [170] further used CRISPR-Cas9 to construct two independent liver-specific TM6SF2 knockout (TM6SF2-LKO) mice and gave them a high-fat diet and found that Tm6LKO mice showed steatosis and decreased VLDL-TG secretion because TM6SF2 deficiency impairs the dynamic remodeling of newborn ApoB in VLDL assembly. In addition, steatosis and fibrosis were alleviated in mice by the phenotypic rescue of Tm6LKO mediated by AAV8. According to the above, TM6SF2 deficiency can affect VLDL-C secretion, and TM6SF2 can also regulate the gut microbiota. If we can combine its dual mechanism of inhibiting lipid accumulation and regulating intestinal microbes, a dual-target drug will be developed, which will greatly alleviate the process of MASLD. Chitosan oligosaccharides and their derivatives, with their unique structural tunability and multi-target activity, complement VLDL in the fields of metabolic regulation and anti-tumor therapy [171]. By using VLDL as a carrier to encapsulate COS derivatives (such as sulfated COS), they target the high expression of LDLR on tumor cells; COS enhances tumor immunity (such as activating T cells), while VLDL provides an energy blockade (inhibiting tumor lipid uptake).

#### 4.1.2. Effects of GLS1 Inhibition and mTORC1 Activation on VLDL Secretion

Jorge Simon et al. [172] found that inhibiting GLS1 can reduce lipid accumulation and steatosis in hepatocytes, restore phospholipid synthesis and methionine circulation in hepatocytes to reduce oxidative stress, and more importantly, improve VLDL assembly and secretion so that triglycerides in the liver can be transported out and reduced.

Kahealani Uehara et al. investigated the effect of mTORC1 activity on NASH by inducing a NASH model using two diets and a TSC1 gene knockout. They found that decreased mTORC1 activity led to decreased phosphatidylcholine synthesis and VLDL-TG secretion, which aggravated inflammation and fibrosis. In contrast, the activation of mTORC1 enhances VLDL-TG secretion and inhibits DNL to reduce TAG accumulation in the liver [173]. In addition, mTORC1 [173] promotes TAG secretion by regulating the key rate-limiting enzyme of PC synthesis, choline phosphocytidine transferase α (CCTα). The results suggest that GLS1 and mTORC1 play an important role in regulating lipid metabolism and VLDL secretion in the liver. By selectively regulating the activity of GLS1 and mTORC1, liver lipid metabolism can be improved and the progression of a fatty liver and fibrosis can be alleviated.

#### 4.1.3. Regulation of VLDL Secretion by HNF4α

Yanyong Xu et al. [174] revealed that HNF4α is a key regulator of liver TG lipolysis activity and fatty acid oxidation by studying a variety of mouse models and diets (including high fat/cholesterol/fructose, HFCF) and adeno-associated virus (AAV)-mediated gene overexpression or knockout techniques. HNF4α may affect TG levels by regulating VLDL secretion [175]. The deletion of HNF4α in the liver has been shown to decrease the expression of MTP and ApoB in the liver [176]. These two proteins play a key role in the secretion of VLDL, and their decreased expression will seriously impair the secretion of VLDL, thus aggravating the occurrence and development of MASLD. It can be seen from the above that HNF4α reduces lipid toxicity by promoting liver lipolysis, fatty acid oxidation, and cholesterol synthesis into bile acids, thereby protecting the liver from the effects of MASLD, while AAV-mediated HNF4α improves liver metabolic function, which provides a theoretical basis for gene therapy strategies. In other words, we can target the up-regulation of HNF4α activity to improve VLDL secretion and slow down the occurrence of MASLD.

#### 4.1.4. Inhibition of MDM2–ApoB Interaction

This study found that the hepatocellular specific knockout of MDM2 protected mice from liver steatosis and inflammation caused by a high-fat, high-cholesterol diet and was accompanied by a significant increase in TG-VLDL secretion. As an E3 ubiquitin ligase, MDM2 targets ApoB for proteasomal degradation through direct protein–protein interactions, resulting in decreased TG-VLDL secretion in hepatocytes. Blocking the MDM2–ApoB interaction by pharmacological means can induce the expression of ApoB and promote the secretion of TG-VLDL, thereby alleviating diet-induced steatohepatitis and fibrosis. MDM2 can affect the secretion of TG-VLDL by regulating the stability of ApoB and can thereby regulate liver lipid metabolism. Inhibiting the MDM2–ApoB interaction not only improves lipid accumulation in the liver but also reduces inflammation and fibrosis. Future studies could further explore the potential of MDM2 as a therapeutic target and develop drugs that target MDM2–ApoB interactions, providing new strategies for the treatment of MASLD [177].

### 4.2. Clinical Exploration of Targeted VLDL Treatment Methods

In the treatment of MASLD, a variety of drugs have demonstrated potential therapeutic effects by regulating the synthesis, secretion, and metabolism of VLDL through different mechanisms. Lomitapide [178], an MTP inhibitor, has been approved for the treatment of familial hypercholesterolemia but has also shown potential therapeutic benefits in MASLD. Metformin indirectly reduces VLDL secretion by improving insulin sensitivity [179,180]. Bile acids (such as fenofibrate) increase fatty acid oxidation and decrease VLDL synthesis by activating PPARα. Ezetimibe can reduce the absorption of cholesterol and improve dyslipidemia. Several antisense oligonucleotides (ASOs) are being investigated to reduce the expression of ApoB, thereby reducing the synthesis of VLDL. Evolocumab and Alirocumab, proprotein converting enzyme subtype 9 (PCSK9) inhibitors [181] that have been approved for the treatment of hypercholesterolemia, have also shown potential effects in MASLD. PCSK9 plays a role as a potential co-chaperone protein in the endoplasmic reticulum. Its interaction with endoplasmic reticulum chaperone protein GRP94 can reduce LDLR degradation mediated by PCSK9 [182]. Another effect is to inhibit the degradation of ApoB in cells through the autophagosome/lysosomal pathway to achieve the effect of treating MAFLD [183,184]. These drugs regulate VLDL synthesis, secretion, and metabolism through different mechanisms, providing a variety of potential intervention strategies for the treatment of MASLD.

### 4.3. VLDL Quantification: Traditional vs. Modern Approaches

#### 4.3.1. Traditional Methods

In clinical practice, traditional methods such as the Friedewald method [185], ultracentrifugation method [186], and chemical precipitation method are usually used to monitor and quantify VLDL levels so as to provide strong support for the diagnosis and treatment of MASLD. As one of them, the Friedewald method is widely used in clinical laboratories to evaluate LDL cholesterol levels. However, this method showed significant limitations when TG levels exceeded 4.5 mmol/L. In this case, the Friedewald method may not be able to provide accurate results, so other, more reliable methods are recommended.

#### 4.3.2. Modern Approaches

At present, the cutting-edge technologies for the quantitative detection of VLDL include the Liposcale test, the stable isotope tracer technique, and direct enzymatic detection. Among them, the Liposcale test [187], as an innovative technology, is realized by means of nuclear magnetic resonance technology. Nuclear magnetic resonance (NMR) spectroscopy can be used to determine lipoprotein subclasses including VLDL. Compared with traditional methods, this technique is more detailed and accurate in evaluating lipoprotein profiles. It is able to identify and quantify different VLDL subfractions, which is particularly important in understanding the vascular risks associated with dyslipidemia. In the process of using the Friedewald formula to accurately calculate LDL cholesterol, a 12 h fast is usually required to perform a blood lipid test to standardize the measurement of TGs [188]. However, recent studies suggest that non-fasting TG levels may also be an important indicator of cardiovascular disease risk [189]. Therefore, the necessity of fasting depends on the specific clinical situation and the test items carried out.

## 5. Conclusions

MASLD has emerged as the predominant cause of chronic liver disease globally, imposing a substantial economic strain on countries worldwide. This condition is a multifaceted disorder with an intricate pathogenesis involving a plethora of factors such as genetics, oxidative stress, insulin resistance, chronic inflammation, and fibrosis. Due to the incomplete understanding of its pathogenesis and progression, no definitive therapeutic solutions exist, and the US Food and Drug Administration (FDA) has sanctioned the use of only one medication for its treatment. On 14 March 2024, the FDA approved Resmetirom as the first treatment for adult patients with NASH that improves liver function by enhancing metabolism and lowering cholesterol and triglyceride levels.

In recent years, numerous studies in the literature have focused on modeling the effects of high-fat and high-sugar diets on obesity and fatty liver disease. Researchers have delved deeply into the mechanisms of oxidative stress, lipid metabolism, and inflammatory pathways in the context of the role of MASLD and the development of adjuvant therapies. Elevated hepatic lipid content results in increased VLDL secretion, which is a primary driver of complex dyslipidemia in MASLD patients. Apart from the recognized contributions of oxidative stress and VLDL output to the development of a fatty liver, we propose a novel perspective that centers on investigating the synthesis, assembly, secretion, and impact of VLDL in response to an impaired diet, thereby identifying potential targets for the treatment of MASLD.

Interventions aimed at reducing VLDL secretion could alleviate dyslipidemia but may inadvertently worsen MASLD, implying that the balance between lipid storage and secretion in hepatocytes is crucial in determining the disease’s outcome. Therefore, a comprehensive understanding of these processes and the factors that influence them will allow us to develop pretreatment and preconditioning strategies through the VLDL pathway to halt the initial and subsequent progression of liver disease.

This pathological damage is characterized by the accumulation of fat in the liver, which can progress to more severe conditions such as hepatitis and cirrhosis if left untreated. The role of VLDL in this process is crucial, as it is responsible for transporting triglycerides from the liver to other parts of the body. When VLDL levels are elevated, however, this transport mechanism is disrupted, leading to the buildup of triglycerides in the liver and the subsequent development of MASLD.

As previously discussed, TM6SF2 gene knockout, inhibition of MDM2–ApoB interaction, HNF4α, GLS1 inhibition, and mTORC1 activation on VLDL lipidization have shown great potential for reducing steatosis by regulating VLDL to affect MASLD. In addition, lifestyle modifications such as weight loss, regular physical activity, and a healthy diet can significantly impact VLDL levels and overall lipid metabolism. These interventions not only improve insulin sensitivity but also enhance the body’s ability to effectively clear VLDL from the bloodstream, thereby reducing the fat burden on the liver.

Although current research has highlighted the significant role of VLDL in MASLD, several limitations remain. Firstly, the specific molecular mechanisms of VLDL synthesis and secretion are not fully understood. The regulatory mechanisms under different pathological conditions require further investigation, as this complexity has been well documented. Secondly, there are pathophysiological differences between various diseases. These differences may limit the generalizability of findings from animal models to human MASLD. This is because mice express ApoBEC-1 and primarily secrete ApoB48, whereas human livers secrete ApoB-100; therefore, when comparing the results of experiments on lipoprotein metabolism (especially the pathways involving VLDL assembly and clearance) between species, greater caution is required. Moreover, the limitations of current detection technologies also impact the accurate monitoring of dynamic changes. Regarding treatment strategies, no highly specific therapeutic targets have been identified yet, and further research is needed to understand the side effects and tolerability of potential drugs. Future research should focus on interdisciplinary collaboration, developing new detection technologies, and personalized treatment strategies to advance the diagnosis and treatment of MASLD. The future management of NAFLD may incorporate VLDL heart regulation and metabolic modulation methods, despite the challenges in developing specific drugs. Of course, besides the assembly and secretion of VLDL, activating the autophagy pathway can reduce lipid accumulation in the liver without significantly increasing plasma lipid levels. Some studies have shown that activating the autophagy pathway can reduce liver lipid accumulation by increasing lipophagy and breaking down lipids into energy through β-oxidation, thereby reducing the occurrence of a fatty liver.

## Figures and Tables

**Figure 1 biomolecules-15-00990-f001:**
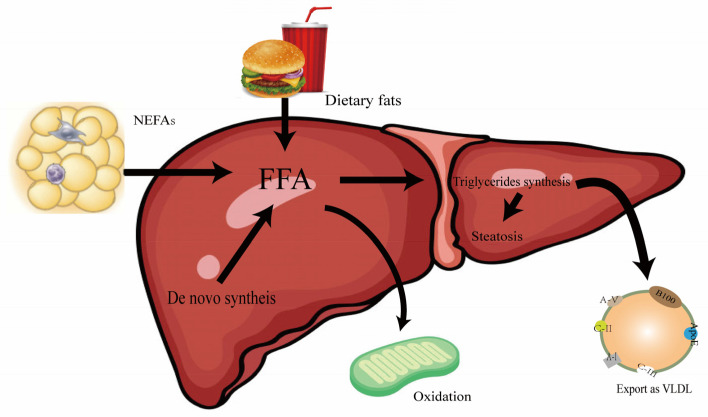
Plot of triglyceride accumulation in the liver. The metabolic process of FFA in the liver and its relationship with dietary fat and NEFAs. FFA is the central substance of fatty acid metabolism, which is derived from the following: dietary intake; de novo fat synthesis; and non-esterified fatty acids. The destination is through the liver mitochondria oxidation, converting to triglycerides, and through the VLDL transport in the liver.

**Figure 2 biomolecules-15-00990-f002:**
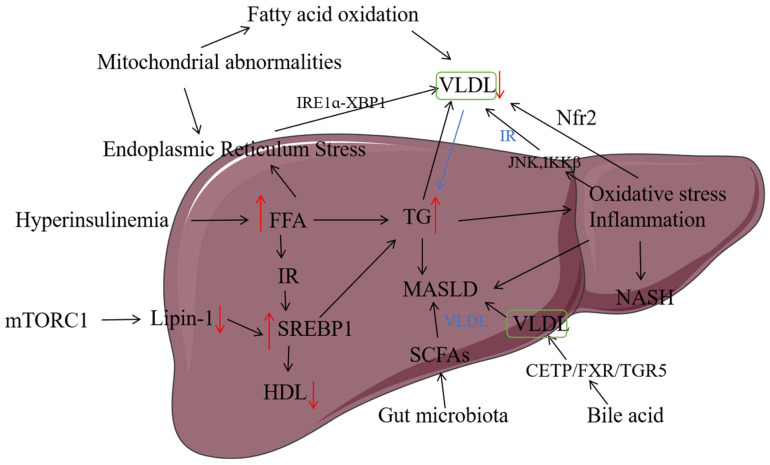
The relationship between the pathogenesis of MASLD and VLDL. VLDL (very low-density lipoprotein); FFA (free fatty acid); IR (insulin resistance); SREBP-1C (Sterol Regulatory Element Binding Protein-1C); HDL (high-density lipoprotein); TGs (triglycerides); SCFAs (short-chain fatty acids); CETP (Cholesteryl Ester Transfer Protein); FXR (Farnesoid X Receptor); TGR5 (TGR5 Bile Acid Receptor); JNK (c-Jun N-terminal kinase); IKKβ (IκB Kinase β); Nrf2 (nuclear factor E2-related factor 2); IRE1α-XBR1 (Inositol-Requiring Enzyme 1 alpha–X-Box Binding Protein 1). Note: ↑ indicates an increase in content, and ↓ indicates a decrease in content.

**Figure 3 biomolecules-15-00990-f003:**
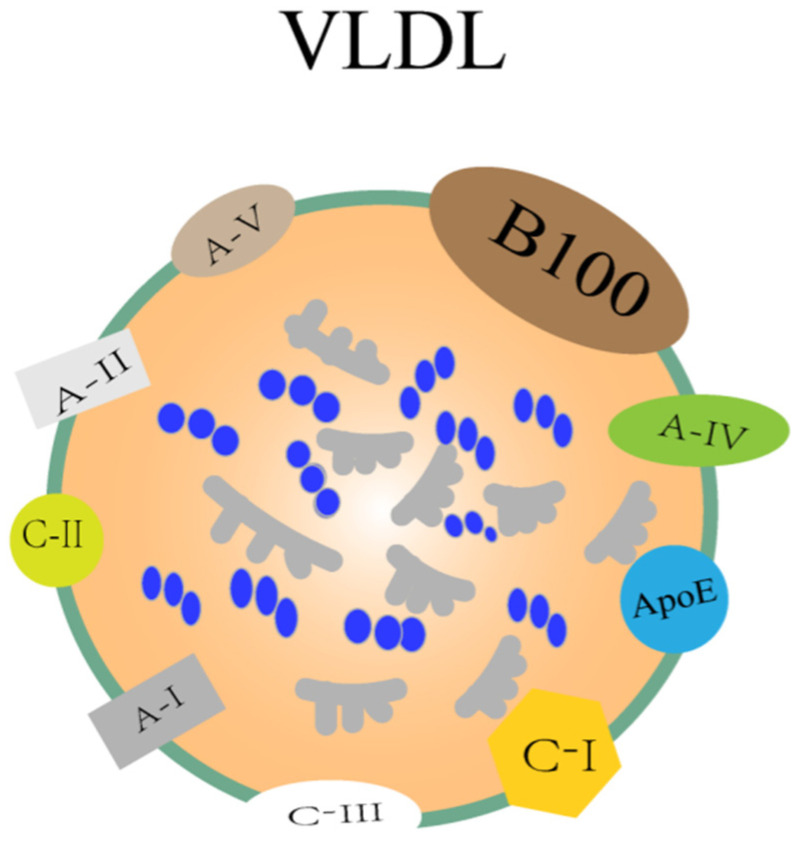
Structure of VLDL. The components of VLDL mainly include TG, cholesterol, cholesterol esters, phospholipids, and proteins. The protein part includes apolipoprotein ApoE, ApoC-I, ApoC-II, ApoC-III, ApoA-I, ApoA-II, ApoA-IV, and ApoA-V.

**Figure 4 biomolecules-15-00990-f004:**
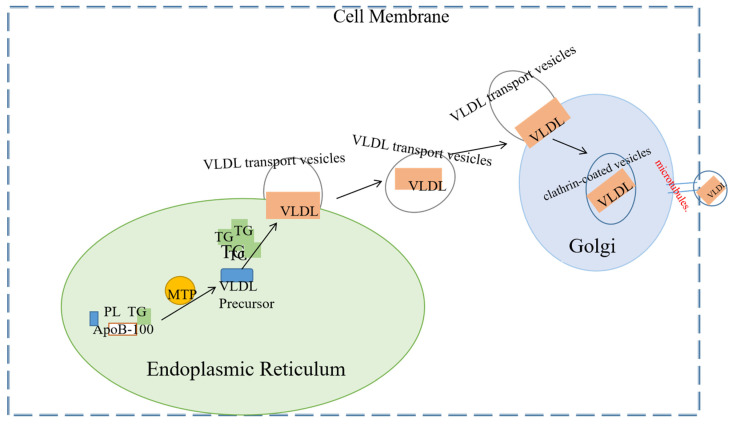
This is a simple diagram of VLDL assembly, secretion, and transport out of the cell. The biosynthesis of VLDL begins in the endoplasmic reticulum (ER), the primary site for protein synthesis and lipid metabolism. Newly synthesized apolipoprotein B-100 (ApoB-100) must undergo a lipidation process mediated by microsomal triglyceride transfer protein (MTP). This process involves the binding of triglycerides (TGs) and phospholipids (PLs) to ApoB-100, forming the precursor of VLDL. The VLDL precursor then binds additional triglycerides (TGs) to form mature VLDL particles. Mature VLDL particles are transported from the endoplasmic reticulum (ER) to the Golgi apparatus via vesicle transport. In the Golgi apparatus, VLDL particles undergo significant modifications, including glycosylation and further lipidation, which ultimately complete their maturation. After these modifications, the mature VLDL particles are packaged into transport vesicles and secreted into the bloodstream via exocytosis.

**Figure 5 biomolecules-15-00990-f005:**
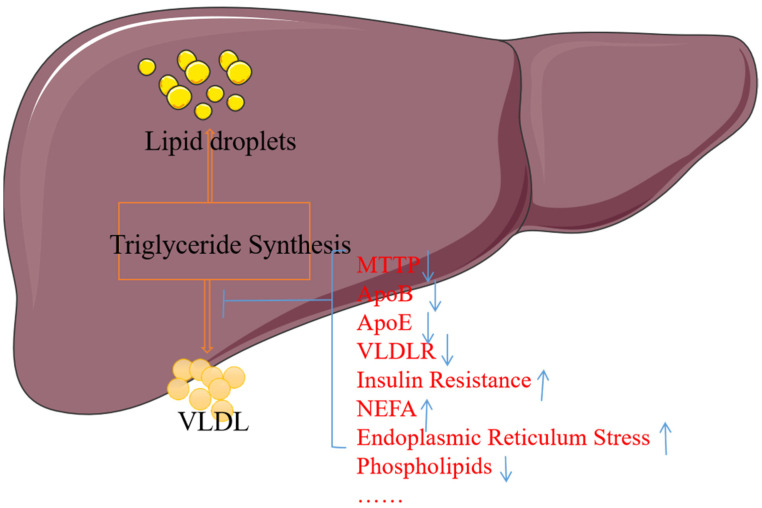
The relationship between steatosis and VLDL secretion. MTTP, ApoB, ApoE, VLDL-R, and phospholipids damaged or missing will affect the synthesis or secretion of VLDL; similarly, insulin resistance, NEFA, or endoplasmic reticulum stress activation or overload will also affect VLDL, leading to lipid accumulation. Note: ↑ indicates an increase in content, and ↓ indicates a decrease in content.

## Data Availability

The original data supporting the conclusions of this article will be provided by the authors upon request.

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
