# Peer review of "Metabolic Dysfunction-Associated Steatotic Liver Disease: From a Very Low-Density Lipoprotein Perspective"

_biomolecules, 2025, doi:10.3390/biom15070990_

Round 1
Reviewer 1 Report (Previous Reviewer 2)
Comments and Suggestions for Authors
The manuscript of Chen et al. is a review on the role VLDL in metabolic-dysfunction associated steatotic liver disease (MASLD). The is a resubmission of a manuscript that has been previously reviewed. The authors have made many changes since the first submission improving its overall quality. The authors have described a complete summary of the many roles of VLDL in liver triglyceride metabolism. Overall, I am happy with the content and the quality of the writing (just a few grammar mistakes). I have a few comments. I am happy that the authors have spent some time discussing the role of CE in VLDL as well as TG. The authors should mention about cholesterol synthesis and regulation, as it relates to VLDL secretion, as well as the role of SOAT1/SOAT2. Line 378: PEMT? Or did the authors mean PERK? Protein kinase RNA-like endoplasmic reticulum kinase? Line 471: hypertriglyceridemia Section 3.3.3: The authors could mention the role of CETP here, especially if discussing relative increase in LDL-CE. Section 4: PCSK9 plays an intracellular chaperone role in internal apoB trafficking. This could be briefly discussed. P.22 abbreviations list: PERK Protein kinase RNA-like endoplasmic reticulum kinase
Comments on the Quality of English LanguageA few mistakes (grammar and spelling) that do not weaken the message of the manuscript. A proper read through should be sufficient to find the last mistakes.
Author Response
Comment 1: The authors should mention about cholesterol synthesis and regulation, as it relates to VLDL secretion, as well as the role of SOAT1/SOAT2.
Response 1: Thank you very much for your valuable comments and suggestions on our article. You mentioned the important content about cholesterol synthesis and regulation and the role of SOAT1/SOAT2 in VLDL secretion, and we fully agree with your views. We have added a small paragraph in the manuscript: The synthesis and regulation of cholesterol are vital for the secretion of VLDL. Cholesterol is mainly produced via the HMG-CoA reductase (HMGCR) pathway, which serves as the key rate-limiting enzyme in this process. Cholesterol synthesis is primarily regulated through feedback mechanisms. When cellular cholesterol levels increase, HMGCR activity is inhibited, reducing cholesterol production. Conversely, when cholesterol levels drop, HMGCR activity rises, enhancing cholesterol synthesis. Additionally, the esterification of cholesterol significantly influences the assembly and secretion of VLDL. Sterol O-acyltransferases 1 and 2 (SOAT1 and SOAT2) play a crucial role in cholesterol esterification. SOAT1 is predominantly located in the endoplasmic reticulum and is responsible for converting free cholesterol into cholesteryl esters, which are then packaged into VLDL particles. SOAT2, on the other hand, is mainly found in the Golgi apparatus and participates in further modification and transport of cholesteryl esters. The activity of SOAT1/SOAT2 directly affects the formation of cholesteryl esters and the efficiency of VLDL secretion. Thank you again for your careful review and valuable comments.(Line 407-420)
Comment 2: PEMT? Or did the authors mean PERK? Protein kinase RNA-like endoplasmic reticulum kinase?
Response 2: Thank you for your careful reading and valuable suggestions on line 378. We have carefully reviewed our manuscript and found that the term "PEPK" actually appears on line 332, not line 378. Upon further research, we learned that PEPK stands for "pyruvate kinase." We regret this typographical error, which we have corrected in line 332 of the manuscript.Thank you again for your careful review and valuable comments.(Line 332)
Comment 3: Line 471: hypertriglyceridemia
Response 3: Thank you very much for pointing out the spelling error in the manuscript. I have corrected the term "hypertriglyceridemia" in the manuscript and have checked the entire text again to ensure that the term is correctly spelled in all places where it appears in the manuscript.Thank you again for your careful review and valuable comments.(Line 458)
Comment 4: The authors could mention the role of CETP here, especially if discussing relative increase in LDL-CE.
Response 4: Thank you very much for your valuable suggestions. As you mentioned, the role of CETP is crucial when discussing the relative increase in LDL-CE. We fully agree with your perspective. Our literature review indicates that CETP is a key regulatory protein involved in the exchange of cholesterol esters (CE) and triglycerides (TG) between plasma lipoproteins. CETP facilitates the transfer of CE from HDL to lipoprotein particles rich in apolipoprotein B (apoB), such as VLDL and LDL, while simultaneously transferring TG from these particles to HDL. This process results in a decrease in the CE content in HDL particles The levels of LDL and VLDL particles have decreased, while the content of CE in these particles has increased. In the revised version of our manuscript, we added a section to address this aspect. We explained how CETP facilitates the transfer of cholesterol esters from HDL to LDL, which aids in understanding the increase in LDL-CE. This process is crucial for understanding the dynamics of lipid metabolism and its impact on cardiovascular disease risk. We believe that this addition will provide a more comprehensive understanding of the mechanisms involved in our research.Thank you again for your careful review and valuable comments.(Line 628-633)
Comment 5: Section 4: PCSK9 plays an intracellular chaperone role in internal apoB trafficking. This could be briefly discussed.
Response 5: Thank you for your suggestion in Section 4 regarding the cellular role of PCSK9 as an intracellular chaperone in internal apolipoprotein transport. We agree that this is an important aspect worth mentioning, and we have added a concise discussion to highlight its significance. Through literature review, we learned that PCSK9 (proprotein convertase subunit 9) plays a crucial role in LDL metabolism. It acts as a chaperone that binds to the LDL receptor (LDLR), targeting its degradation, thereby indirectly increasing LDL levels. Studies have shown that PCSK9 functions as a potential co-chaperone in the endoplasmic reticulum, It interacts with the endoplasmic reticulum (ER) chaperone protein GRP94, which can reduce the degradation of LDL receptor-like (LDLR) mediated by PCSK9 and increase the protein abundance of GRP94. Additionally, PCSK9 in the ER can act as a chaperone, facilitating the transport of immature ER-resident LDLRs to the cell surface (Reference 5). Apolipoprotein B (apoB), the primary structural protein of VLDL and LDL particles, suggests that PCSK9's role as a chaperone in LDLRs might indirectly affect the intracellular transport of apoB. However, literature review reveals that the folding and stability of apoB The qualitative aspects primarily depend on the molecular chaperones and quality control mechanisms within the endoplasmic reticulum (ER) (Reference 2). Research has confirmed that PCSK9 inhibits the degradation of apoB through the autophagosome/lysosome pathway (References 1, 4). Additionally, studies have shown that PCSK9 interacts with apoB and regulates its production, indicating a second function of PCSK9 as a chaperone (Reference 3). Furthermore, PCSK9 interacts with ApoB and regulates its production, suggesting that PCSK9 may indirectly influence ApoB by aiding its folding and stabilization within the cell The intracellular transport of PCSK9 (Reference 6). PCSK9 directly interacts with apolipoprotein B (ApoB), including the primary structural lipoproteins ApoB100 in LDL and VLDL, and ApoB48 in chylomicrons (CM), thereby reducing degradation, increasing ApoB secretion, and enhancing the stability of ApoB protein. (Reference 7)
Therefore, we have added this discussion to the manuscript: PCSK9 (Proprotein Conversion Enzyme Subtype 9) functions as a potential co-chaperone in the endoplasmic reticulum. It interacts with the endoplasmic reticulum chaperone protein GRP94, which can reduce PCSK9-mediated LDLL degradation. Another mechanism involves inhibiting the degradation of apoB within cells through the autophagy/lysosome pathway, thereby treating MAFLD.Thank you again for your careful review and valuable comments.(Line 787-791)
References 1: Sun, Hua et al. “Proprotein convertase subtilisin/kexin type 9 interacts with apolipoprotein B and prevents its intracellular degradation, irrespective of the low-density lipoprotein receptor.” Arteriosclerosis, thrombosis, and vascular biology vol. 32,7 (2012): 1585-95. doi:10.1161/ATVBAHA.112.250043
Reference 2:Kumari, Deepa, and Jeffrey L Brodsky. “The Targeting of Native Proteins to the Endoplasmic Reticulum-Associated Degradation (ERAD) Pathway: An Expanding Repertoire of Regulated Substrates.” Biomolecules vol. 11,8 1185. 11 Aug. 2021.
Reference 3:https://www.ahajournals.org/doi/abs/10.1161/circ.122.suppl_21.A18351
Reference 4:Sun, Hua et al. “PCSK9 deficiency reduces atherosclerosis, apolipoprotein B secretion, and endothelial dysfunction.” Journal of lipid research vol. 59,2 (2018): 207-223. doi:10.1194/jlr.M078360
Reference 5:Lebeau, Paul F et al. “The Emerging Roles of Intracellular PCSK9 and Their Implications in Endoplasmic Reticulum Stress and Metabolic Diseases.” Metabolites vol. 12,3 215. 26 Feb. 2022, doi:10.3390/metabo12030215
Reference 6:Strøm, Thea Bismo et al. “PCSK9 acts as a chaperone for the LDL receptor in the endoplasmic reticulum.” The Biochemical journal vol. 457,1 (2014): 99-105.
Reference 7:Rashid, Shirya et al. “Proprotein convertase subtilisin kexin type 9 promotes intestinal overproduction of triglyceride-rich apolipoprotein B lipoproteins through both low-density lipoprotein receptor-dependent and -independent mechanisms.” Circulation vol. 130,5 (2014): 431-41. doi:10.1161/CIRCULATIONAHA.113.006720IF: 38.6 Q1
Comment 6: P.22 abbreviations list: PERK Protein kinase RNA-like endoplasmic reticulum kinase
Response 6: Thank you for your feedback on the abbreviations list on page 22. The abbreviation "PERK" stands for "Protein Kinase RNA-like Endoplasmic Reticulum Kinase," which we mistakenly wrote as "REPK" earlier. We apologize for this error and will make the necessary corrections in the manuscript. Thank you again for your careful review and valuable suggestions.(Line 901)
Comment 7: A few mistakes (grammar and spelling) that do not weaken the message of the manuscript. A proper read through should be sufficient to find the last mistakes.
Response 7: Thank you for pointing out some minor grammatical and spelling errors in our manuscript. We have taken your suggestions seriously and conducted a thorough proofreading process. During this process, we identified and corrected several word errors that were overlooked in the initial draft. We have also implemented additional checks to ensure similar errors do not occur in the future. We believe this further improves the clarity and readability of our work.We appreciate your patience and understanding. It is shown in orange in the manuscript.
Reviewer 2 Report (New Reviewer)
Comments and Suggestions for Authors
Strengths of the review:
- This is a very interesting and comprehensive review from a biochemical perspective, which addresses in detail MASLD, a multifactorial disease that is increasingly increasing worldwide and primarily affects the liver.
- The authors of the review present the mechanisms of oxidative stress, lipid metabolism, and inflammatory pathways in the context of the role of MASLD and the development of adjuvant therapies.
- In this regard, they propose alternatives focused on the synthesis, assembly, and secretion of VLDL in response to a poor diet, thus identifying potential targets for the treatment of MASLD.
- With the mechanisms thoroughly explained, pretreatment and preconditioning strategies can be developed through the VLDL pathway to halt the initial and subsequent progression of liver disease.
- It is noteworthy that inactivation of the TM6SF2 gene, and inhibition of the MDM2-ApoB interaction, as well as inhibition of HNF4α and GLS1, and subsequent activation of mTORC1 in VLDL lipidation have great potential to reduce steatosis by regulating VLDL to affect MASLD.
- On the other hand, the limitations they propose are very interesting, among which the presence of specific molecular mechanisms, the regulatory mechanisms in different pathological conditions, stands out. Furthermore, the pathophysiological differences that exist between various diseases limit the findings from animal models to human MASLD.
- The authors propose that management of MASLD requires interdisciplinary collaboration, involving the development of new screening technologies and personalized treatment strategies to advance the diagnosis and treatment of MASLD.
- Regarding the references that support this manuscript, they are adequate and relevant for the development of the topic.
Weakness:
The only weakness I detect in this review is that it did not address the most current laboratory tests for quantifying VLDL, such as the Liposcale test, and that it did not compare the virtues of this technique in contrast to conventional tests that are determined in the clinical laboratory where the Fridewall method is used, in addition to discussing when it is necessary to fast for 12 hours and when it is not.

Author Response
Comment 1: It is suggested to change intestinal flora to gut microbiota
Response 1: Thank you very much for your valuable feedback on our paper. Following your suggestions, we have revised 'intestinal flora' to 'gut microbiota.' We recognize that 'gut microbiota' is a more scientific and accurate term, as it encompasses a broader range of microorganisms in the gut, including bacteria, fungi, viruses, and more than just bacteria. This revision will help standardize the use of professional terms in our paper and align with current academic trends and consensus. Thank you again for your professional guidance. (Line 191/335/336/347/350)
Comment 2: it did not address the most current laboratory tests for quantifying VLDL, such as the Liposcale test, and that it did not compare the virtues of this technique in contrast to conventional tests that are determined in the clinical laboratory where the Fridewall method is used, in addition to discussing when it is necessary to fast for 12 hours and when it is not.
Response 2: Thank you for your valuable feedback. We appreciate and will incorporate your suggestions into our manuscript for further improvements. Therefore, the discussion on VLDL detection methods can be considered as a supplement, placed after the section on 4.2 Clinical Exploration of Targeted VLDL Therapies and before the conclusion. We have added a new section to 4.3: VLDL Quantification: Traditional Methods vs. Modern Methods, to highlight its clinical significance. The measurement of VLDL levels in the trial is aimed at supporting the diagnosis and treatment of MASLD. Thank you again for your professional guidance. (Lines 794-813)
This manuscript is a resubmission of an earlier submission. The following is a list of the peer review reports and author responses from that submission.
Round 1
Reviewer 1 Report
Comments and Suggestions for Authors
[Biomolecules] Manuscript ID: biomolecules-3628316 - Review Report
In this review, the authors evaluate the effects of VLDL synthesis and metabolism on MAFLD, including changes in VLDL structure and composition, the biosynthesis of VLDL, and the mechanism of VLDL damage, to elucidate the intricate crosstalk between MAFLD and VLDL and provide a new perspective and method for the prevention and treatment of related diseases. Below are my comments and suggestions on your work.
Please introduce abbreviations at their first occurrence in the text and use them thereafter. For instance, terms like MAFLD, FATP, and LCPUFA are introduced multiple times, or you use them without prior definition.
Please review the citation in the manuscript. In several instances, authors' first and last names are provided (e.g., "Alaric Falcon et al."), which may not align with the journal's citation style.
Line 44–45. The sentence "Alaric Falcon et al. [2] showed that FATP2 knock-out reduced fatty acid uptake and ameliorated high-fat diet-induced hepatocyte degeneration", please write it more clearly.
Introduce the abbreviation for de novo lipogenesis (DNL) at its first mention in the text.
In Lines 53–54, when discussing the desaturation and elongation of fatty acids, please specify the enzymes involved in these processes. Also, point out that these pathways lead to the synthesis of monounsaturated and polyunsaturated fatty acids.
In line 70, replace the term "encapsulated" with a more appropriate term that describes the synthesis of the VDLD particle.
In Lines 110–120, the same abbreviation is introduced multiple times. Please, correct it.
Lines from 138 to 142 are not clear, please improve them.
Lines from 150 to 158 are not clear, please improve them.
In Lines 178–194, you can arrange the four lipid metabolism pathways in numerical order for better understanding.
Please provide appropriate references that confirm the statement: "Elevated oxidative stress leads to progressive hepatocyte death." Line 235-236
Please specify information on the role of oleic acid in HepG2 cells for better understanding.
Please, define "endoplasmic reticulum stress" at the beginning as a due to a better understanding.
Rephrase the sentence in Lines 307–309 to be clear and precise.
Please specify a clearer explanation of the content in Lines 327–334.
Line 359: Specify that "this lipid" refers to VLDL particles in this part.
In the part about VLDL particles before mentioning choline, list all classes of phospholipids to provide, then move on to choline. Also, when discussing VLDL particles, mention all the molecules that are included in the composition of VLDL. In this part, it looks like phosphatidylcholine is the only phospholipid that is found in the VLDL particle. Later, you mention sphingomyelin but do not mention phosphatidylethanolamine.
Free fatty acids are essential for triglyceride synthesis, which also requires glycerol. Therefore, instead of using the term "converted," use a more precise term that precisely indicates the biochemical process.
Reviewer 2 Report
Comments and Suggestions for Authors
The manuscript of Chen et al. describes the role of VLDL (and its accompanying lipids) in the formation and progression of non-alcoholic fatty liver disease (NAFLD)/metabolic dysfunction associated fatty liver disease (MAFLD)/metabolic dysfunction associated steatotic liver disease (MASLD). The authors have described the many roles of VLDL and the many factors affecting its production, metabolism and the issues related to its impaired secretion. I find the topic important, although there are only a few novel insights compared to recently published reviews on the same topic. Grammar is problematic, word choice is awkward. There are many considerations for this review to be improved. As a passive observer, I have seen the name of NAFLD change to MAFLD and now MASLD. I guess I don't really mind which term people use, just as long as it is consistent. If MASLD is the most up to date term, widely accepted, then I would recommend that the authors adopt this term instead. If there is no consensus, or the authors have a strong preference, then I have no objection. Line 108: When referring to VLDL-TG, I assumed that the authors are referring to the TG in VLDL. In the context of the sentence, however, I guess the authors are referring to a TG-enriched VLDL? Small dense LDL (sdLDL) is composed primarily of CE and is thought to be more atherogenic (especially at increased numbers) because the same [LDL-C] is distributed in more particles, which is why some clinicians also measure apoB concentration, not just [LDL-C]. So, the sentence doesn't really make sense how a TG-enriched VLDL could produce more CE-enriched LDL. Please clarify. Also Line 219. Line 518. Line 116-121: run-on sentence Line 151: I am unfamiliar with the "second strike" hypothesis, but am familiar with a "Two-hit" or "Multi-hit" hypothesis. Again I would urge the authors to be consistent with the current literature (whatever it is, as I may simply not be familiar with the term). Figure 2: oxidative stress is misspelled. Inositol-requiring transmembrane kinase/endoribonuclease 1α (IRE1α) Section 2: Some mention of transient steatotic states in the liver and the role of mTORC1 and Lipin-1 should be mentioned (https://www.jci.org/articles/view/96036; https://www.nature.com/articles/s41418-025-01507-6; https://www.nature.com/articles/s44324-024-00018-1) Line 280: PERK Line 366-367: PEMT KO leads to less VLDL secretion, so the methylation pathway of PC synthesis may be more important for that? (https://pubmed.ncbi.nlm.nih.gov/19520976/). You mentioned this later (lines 565-569). ​Line 375: VLDL is listed twice. Line 380: I would prefer if the authors did not use etc. here and instead just list all VLDL associated apolipoproteins. Line 381, 382: The role of apoC-III needs to be expanded. Intracellular apoC-III promotes VLDL secretion, plasma apoC-III prevents LPL/apoC-II mediated lipolysis of VLDL. Both of these mechanisms can have an effect on MAFLD. Line 391: conformated is not a word. Line 393: CLDL Line 394: HTG usually refers to hypertriglyceridemia As a more general point, free cholesterol and cholesteryl ester (CE) also accumulates in NAFLD, and might actually be more pathogenic. While the majority of steatosis is derived from TG accumulation (depending on the condition), the authors have not commented on any role of accumulated FC or CE. Figure 4: Text in the figure is misprinted in my version of the manuscript Section 3.2: It is unreasonable to expect you to go into great detail about all the factors involved in VLDL assembly and secretion, but it would be helpful to refer to a comprehensive review on the subject (for example, https://pubmed.ncbi.nlm.nih.gov/38236950/) Line 466: I think you mean "nascent" Line 527: The authors refer to an old paper that may not provide the full story. ApoE is involved in uptake of dietary lipids from chylomicrons and chylomicron remnants. This would increase delivery of dietary lipids to the liver, where increased VLDL secretion occurs. So it is an issue of supply that drives VLDL secretion. This paper does not show that apoE promotes VLDL secretion directly. Perhaps another paper does. I would ask the authors to more accurately describe these results. ApoE also binds to other receptors including LRP1 and HSPGs. Line 546: immortalized Line 549: lncRNA not IncRNA When describing lipoprotein metabolism, it is important to describe that mouse experiments (both liver cells and animals) produce apoB48 from the liver (as the mouse liver has apoBEC1). For this reason, it is not always easy to compare experimental results from mouse systems to humans. Line 603: "impures"???? Maybe impairs? I would caution the authors to not be so quick to promote TG emptying of the liver by increasing TG-enriched VLDL secretion. These TGs and fatty acids have to go somewhere. TG accumulation in adipose tissue can eventually lead to inflammation. TG accumulation in muscle leads to insulin resistance. You might be curing the MAFLD but exacerbating other conditions. Rather, the best way to reduce fatty liver is to destroy the lipids, by increasing lipophagy and b-oxidation. Insights into ways to increase lipophagy and b-oxidation should be explored in this review. I would ask the authors, in rebuttal, to describe to me what is novel in their review compared to other recent reviews on the topic.
Comments on the Quality of English LanguageLots of mistakes. It looks like a translation program was used and many translations are incorrect. A first language English speaker should proofread this manuscript.